# A Fruit-Expressed MYB Transcription Factor Regulates Anthocyanin Biosynthesis in *Atropa belladonna*

**DOI:** 10.3390/ijms25094963

**Published:** 2024-05-02

**Authors:** Xiaoqiang Liu, Tengfei Zhao, Lina Yuan, Fei Qiu, Yueli Tang, Dan Li, Fangyuan Zhang, Lingjiang Zeng, Chunxian Yang, Mohammad Mahmoud Nagdy, Zun Lai Lai Htun, Xiaozhong Lan, Min Chen, Zhihua Liao, Yan Li

**Affiliations:** 1Integrative Science Center of Germplasm Creation in Western China (CHONGQING) Science City, School of Life Sciences, Southwest University, Chongqing 400715, China; liuxq@swu.edu.cn (X.L.); tengfeizhao@swu.edu.cn (T.Z.); 17784728625@163.com (L.Y.); feiqiu01@swu.edu.cn (F.Q.); tangyueli_999@163.com (Y.T.); lidans10@126.com (D.L.); fyzhang@swu.edu.cn (F.Z.); zengling@swu.edu.cn (L.Z.); yangchunxian@163.com (C.Y.); zunlailai@gmail.com (Z.L.L.H.); 2Integrative Science Center of Germplasm Creation in Western China (CHONGQING) Science City, College of Pharmaceutical Sciences, Southwest University, Chongqing 400715, China; muhammadnagdy@hotmail.com (M.M.N.); mminchen@swu.edu.cn (M.C.); 3Department of Medicinal and Aromatic Plants Research, National Research Centre, Cairo 12311, Egypt; 4Department of Botany, University of Magway, Magway 04012, Myanmar; 5The Provincial and Ministerial Co-founded Collaborative Innovation Center for R & D in Tibet Characteristic Agricultural and Animal Husbandry Resources, The Center for Xizang Chinese (Tibetan) Medicine Resource, Xizang Agriculture and Animal Husbandry University, Nyingchi 860000, China; lanxiaozhong@163.com

**Keywords:** *Atropa belladonna*, AbMYB1, transcription factor, anthocyanin biosynthesis, regulation

## Abstract

Anthocyanins are water-soluble flavonoid pigments that play a crucial role in plant growth and metabolism. They serve as attractants for animals by providing plants with red, blue, and purple pigments, facilitating pollination and seed dispersal. The fruits of solanaceous plants, tomato (*Solanum lycopersicum*) and eggplant (*Solanum melongena*), primarily accumulate anthocyanins in the fruit peels, while the ripe fruits of *Atropa belladonna* (*Ab*) have a dark purple flesh due to anthocyanin accumulation. In this study, an R2R3-MYB transcription factor (TF), AbMYB1, was identified through association analysis of gene expression and anthocyanin accumulation in different tissues of *A. belladonna*. Its role in regulating anthocyanin biosynthesis was investigated through gene overexpression and RNA interference (RNAi). Overexpression of *AbMYB1* significantly enhanced the expression of anthocyanin biosynthesis genes, such as *AbF3H*, *AbF3*′*5*′*H*, *AbDFR*, *AbANS*, and *Ab3GT*, leading to increased anthocyanin production. Conversely, RNAi-mediated suppression of *AbMYB1* resulted in decreased expression of most anthocyanin biosynthesis genes, as well as reduced anthocyanin contents in *A. belladonna*. Overall, AbMYB1 was identified as a fruit-expressed R2R3-MYB TF that positively regulated anthocyanin biosynthesis in *A. belladonna*. This study provides valuable insights into the regulation of anthocyanin biosynthesis in Solanaceae plants, laying the foundation for understanding anthocyanin accumulation especially in the whole fruits of solanaceous plants.

## 1. Introduction

Anthocyanin, a kind of water-soluble flavonoid pigment substance, is the general name of glycosylated derivatives formed by the combination of anthocyanidins and monosaccharides through glucoside bonds. Anthocyanins play a very important role in the physiological processing of plants, for instance, attracting animals to pollinate, and spreading seeds by providing plants with red, blue, and purple pigments [1]. Due to their various colors and antioxidant capacity, anthocyanins are often used as a color matching agent and preservative in the food industry. In addition, they are also used in anti-aging, antioxidant, and whitening skincare products in the cosmetics industry, and in treating cardiovascular diseases in the pharmaceutical industry. Anthocyanin biosynthesis is regulated by transcription factors (TFs). Pelletier et al. divided the key enzyme genes involved in the flavonoid biosynthesis pathway into two categories: early biosynthesis genes (EBGs) and late biosynthesis genes (LBGs). EBGs in the anthocyanin biosynthesis pathway are key enzyme genes involved in the synthesis of common precursors of flavonoid substances, including *chalcone synthase* (*CHS*), *chalcone isomerase* (*CHI*), *flavanone 3-hydroxylase* (*F3H*), *flavonoid 3*′*-hydroxylase* (*F3*′*H*), and flavonoid 3′, 5′-hydroxylase (*F3*′*5*′*H*). LBGs refer to key enzyme genes specific to the anthocyanin biosynthesis pathway, mainly including *dihydroflavonol 4-reductase* (*DFR*), *anthocyanin synthase* (*ANS*), and *flavonoid 3-O-glucosyltransferase* (*3GT*) [2]. In general, the MBW protein complexes (MYB-bHLH-WD40) regulate LBGs in the anthocyanin pathway, while the MBW-independent MYB TFs mainly regulate EBGs in the anthocyanin pathway [3]. The MYB TFs are the largest TF family involved in the regulation of biological metabolism in plants [4], and the highly conserved functional DNA-binding domain present at the MYB N-terminal is usually composed of 1–3 repeated sequences (R1, R2, and R3). Previous studies have shown that the regulatory activity of MYB TF is the main factor responsible for the variety of plant coloration patterns in nature, among which the R2R3-MYB transcription factor is an important transcription factor regulating anthocyanin biosynthesis in plants [5]. In *Malus domestica* (*Md*), MdMYB1 can directly bind to the promoter of *MdGSTF6* to promote the transfer of anthocyanin to a vacuole [6]; depending on the co-expression of *MdbHLH3* and *MdbHLH33*, *MdMYB10* induces efficient anthocyanin biosynthesis in transient assays [7]. In *Oryza sativa* (rice), the MBW complex formed by R2R3-MYB can activate the expression of all the structural genes involved in anthocyanin synthesis in leaves [8], while in pomegranate, R2R3-PgMYB controls the synthesis of coloration through UFGT [9]. A similar effect of R2R3-MYB has been found in plants such as pears (*Pyrus ussuriensis*) and peaches (*Prunus persica*) [10,11]. Unlike *O. sativa*, *Zea mays* (corn), *Malus pumila* (apple), and *P. persica*, where anthocyanins accumulate mainly in the seed coat or fruit peels, or Solanaceous plants eggplant (*Solanum melongena*) and tomato (*Solanum lycopersicum*), where anthocyanins are not present in the fruit or only present in the fruit peel [12,13], *Atropa belladonna* ripe fruits also have a dark purple flesh.

Solanaceous plants are important sources of food and natural medicine. *A. belladonna*, a member of the belladonna genus within the Solanaceae family, is a valuable medicinal plant used for producing anticholinergic drugs, including hyoscyamine and scopolamine. When fully ripe, the fruits of *A. belladonna* exhibit a purplish-black color and have dark purple juice. In belladonna, both the peels and flesh of the fruit are rich in anthocyanins, reaching maximum abundance during the late ripening stage, and this is very rare in other Solanaceae plants. However, not much research has been carried out on the regulation of anthocyanins biosynthesis in belladonna genus fruits. In this study, an R2R3-MYB TF (AbMYB1) highly expressed in fruits was identified based on the association analysis of gene expression and anthocyanin accumulation in different tissues of *A. belladonna*. The influence of *AbMYB1* on anthocyanin synthesis in *A. belladonna* fruits was investigated through gene overexpression and RNAi, providing evidence for its regulatory role in anthocyanin biosynthesis. This research serves as a foundation for further studies on the regulation of anthocyanin biosynthesis, especially in the fruits of solanaceous plants, and provides a theoretical basis and biological components for future nutritional fortification of Solanaceae fruits with anthocyanins.

## 2. Results

### 2.1. Expression Pattern of AbMYB1 and Anthocyanin Biosynthesis Genes

Based on the gene expression abundance in each organ, the genes with the highest expression in fruits were selected, and the genes expressed only in flowers and fruits and other organs enriched with anthocyanin were preferentially selected. Eight pathway genes involved in anthocyanin biosynthesis were screened based on expression differences, and the screening results are shown in Appendix A. According to the expression abundance of genes in various tissues in the transcription database of *A. belladonna*, the genes in Appendix A and MYB transcription factors in *A. belladonna* were clustered to generate heat maps, and an MYB transcription factor named AbMYB1, whose expression pattern was similar to that of the anthocyanin biosynthesis pathway gene in *A. belladonna*, was identified. Its sequence number in the *A. belladonna* transcriptome is aba_locus_4678_iso_1_len_1035_ver_2. The cluster analysis results showed that anthocyanin biosynthesis pathway genes had high expression abundance in ripe fruits, and most of the pathway genes were also highly expressed in mature seeds, flowers, and green fruits, while very low expression levels were found in roots, stems, leaves, and other parts (Figure 1A). The expression pattern of *AbMYB1* in the roots, stems, leaves, flowers, green fruits, and ripe fruits of wild-type *A. belladonna* was further analyzed. *AbMYB1* possessed the highest expression level in ripe fruits, followed by green fruits, and flowers, while the expression level of *AbMYB1* in roots, stems, and leaves was extremely low (Figure 1B). The results were consistent with the *AbMYB1* digital expression spectrum, which suggested that *AbMYB1* is a tissue-specific transcription factor.

### 2.2. Isolation and Phylogenetic Analyses of AbMYB1 Gene

The coding region of *AbMYB1* is 762 bp in length and encodes 254 amino acids (Appendix A). The amino acid sequences of the MYB transcription factors with known functions were downloaded from NCBI (https://www.ncbi.nlm.nih.gov/, accessed on 23 April 2024) and TAIR (https://www.arabidopsis.org/, accessed on 23 April 2024) websites. A phylogenetic tree was constructed using the Neighbor-Joining (NJ) method and MEGA 7.0 software. The results revealed that AbMYB1 was closely related to SnMYB, followed by PhDPL (Appendix A). Homologous similarity between AbMYB1 and MYB transcription factors involved in anthocyanin biosynthesis regulation in *Arabidopsis thaliana* (*At*), *Solanum nigrum* (*Sn*), and *Petunia hybrida* (*Ph*) was analyzed using Vector NTI Suite 8.0 software. The comparison showed that AbMYB1 shared a high degree of similarity in amino acid sequence with these transcription factors. Additionally, both AbMYB1 and the MYB transcription factors from other species possessed the characteristic structural features of R2R3-MYB transcription factors, including conserved R2 and R3 domains (Figure 2). These findings suggested that AbMYB1 was evolutionarily related to SnMYB and PhDPL, and shared sequence similarity and structural characteristics with MYB transcription factors involved in anthocyanin biosynthesis regulation in other plant species.

### 2.3. Expression Profiles of Anthocyanin Biosynthesis Genes in Wild-Type A. belladonna Fruits at Different Developmental Stages

During the development of the wild-type *A. belladonna* fruit, the accumulation of anthocyanin caused the fruit (flesh and peels) to change color from green to dark purple. In the young fruit stage, both the flesh and peels exhibited a pale green color, indicating a low level of anthocyanin accumulation. As the fruit progressed to the expansion stage, the flesh and peels turned dark green, suggesting an increase in anthocyanin content. At the breaking stage, the peels of the fruit appeared purple, while the flesh remained green, indicating a concentration of anthocyanins in the peels. In the purple fruit stage, both the flesh and peels exhibited a purple color, indicating a significant accumulation of anthocyanins in these tissues. However, the septum, which separates the fruit compartments, was not completely colored, suggesting a lower level of anthocyanin accumulation in this region. Finally, in the ripe fruit stage, the entire fruit turned purple-black, representing the maximum accumulation of anthocyanins (Figure 3).

Fruits of *A. belladonna* at five different developmental stages were selected to investigate the changes in expression levels of anthocyanin biosynthesis pathway genes and *AbMYB1*. Specifically, the expression levels of *AbCHS*, *AbCHI*, *AbF3H*, *AbF3*′*H*, *AbF3*′*5*′*H*, *AbDFR*, and *AbCHS* showed a consistent pattern with *AbMYB1*. They exhibited low expression levels in the young fruit stage, gradually decreasing to the lowest level during the expansion stage. Subsequently, a sharp increase in expression was observed from the expansion stage to the breaking stage, reaching the maximum at the breaking stage. Following the breaking stage, the expression levels gradually decreased from the breaking stage to the ripe fruit stage. In contrast, the expression level of *Ab3GT*, another gene involved in anthocyanin biosynthesis, showed a different pattern. It displayed low expression in the young fruit stage, followed by a decrease from the breaking stage to the ripe fruit stage. Notably, two inflection points were observed in the expression pattern of *Ab3GT*, occurring during the expansion stage and the purple fruit stage, where the expression levels were higher compared to the adjacent stages (Figure 4).

These findings indicated that the expression of anthocyanin biosynthesis genes is closely related to that of *AbMYB1* during *A. belladonna* fruit development. The highest expression levels were observed in the breaking stage, which coincides with the peak of anthocyanin accumulation. This suggested that *AbMYB1* and the analyzed genes play crucial roles in anthocyanin biosynthesis in *A. belladonna* fruits, contributing to the observed color changes throughout fruit development.

### 2.4. Anthocyanin Accumulation in Transgenic A. belladonna Plants

*A. tumefaciens* EHA105 carrying either the plasmid pBI121-*AbMYB1* or pBin19-i*AbMYB1* was employed for the leaf transformation of *A. belladonna*. Following co-cultivation, selective culture, and bud induction culture, the resulting buds (2–3 cm in length) were transferred to a rooting medium for further cultivation to obtain regenerated plants (Appendix A). After one month of cultivation in rooting medium, the regenerated plants were transplanted into a cultivation medium to grow until the plants were around 15 cm tall. *A. belladonna* leaves were taken to extract DNA. Using the extracted DNA as a template, 35S and specific downstream primers were used as PCR detection primers. Three *AbMYB1*-overexpression transgenic lines, named *OE-1*, *OE-2,* and *OE-3,* respectively, and three *AbMYB1*-interfering (RNAi) transgenic lines, named *RI-1*, *RI-2,* and *RI-3,* respectively, (Appendix A) were obtained. *AbMYB1* is mainly expressed in flowers and fruits in wild-type *A. belladonna*; therefore, the expression level of *AbMYB1* in the flowers of the transgenic lines was detected. Compared to the wild type, the expression level of *AbMYB1* was significantly increased in flowers of the *AbMYB1*-overexpression transgenic lines, while it was significantly decreased in flowers of the *AbMYB1*-RNAi transgenic lines (Appendix A), and it was the same with the content of anthocyanin in their flowers, respectively (Figure 5). Unlike no expression in the leaves of the wild type, the expression level of *AbMYB1* in the leaves of the *AbMYB1*-overexpression transgenic lines was significantly increased, leading to the increase in expression levels of most of the anthocyanin biosynthetic pathway genes such as *AbF3H*, *AbF3*′*5*′*H*, *AbDFR*, *AbANS,* and *Ab3GT* (Appendix A), which caused the content of anthocyanin to increase (Appendix A).

Fruit phenotypes of wild-type, *AbMYB1*-overexpression transgenic, and *AbMYB1*-RNAi transgenic *A. belladonna* were surveyed at different developmental stages. Fruits of the wild type still stayed green at 15 days after setting, and turned to purple at around 20 days after setting, and turned to purple black at 25 days after setting when ripening. There was no significant difference between the fruits of the *AbMYB1*-RNAi transgenic lines before the expansion stage (around 15 days) and those of the wild type, but the fruits became lighter green at around 20 days from setting, and showed light yellow ripening at 25 days from fruit setting, which was different from the wild type. In contrast, the fruit peels and sepals of the *AbMYB1*-overexpression transgenic lines showed lilac purple shortly after fruit setting, and the color of the fruit became discernably darker at around 13 days after fruit setting (Figure 6A). Both the peels and flesh of wild-type *A. belladonna* ripe fruits (25 days after fruit setting) showed purplish black, and ripe fruits of the *AbMYB1*-overexpression transgenic lines also showed purplish black, with a deeper color. However, the fruit’s peels and flesh in the *AbMYB1*-RNAi transgenic lines remained light yellow at 25 days after fruit setting. Anthocyanin was extracted from the ripe fruits (25 days after setting) and the content was measured. The anthocyanin content was significantly increased in the *AbMYB1*-overexpression transgenic line fruits, while it significantly decreased to hardly any in the *AbMYB1*-RNAi transgenic line fruits (Figure 6B,C).

### 2.5. Expression Profiles of Related Genes in Transgenic A. belladonna Fruits

Based on the significant change in fruits color of *AbMYB1*-overexpression transgenic plants at fruit setting for 13 days, expression profiles of anthocyanin biosynthesis related genes from different lines were analyzed to explain this phenomenon at the molecular level. The expression levels of genes related to anthocyanin biosynthesis including *AbMYB1*, *AbCHS*, *AbCHI*, *AbF3H*, *AbF3*′*5*′*H*, *AbDFR*, *AbANS* and *Ab3GT* in the fruits of *A. belladonna* were significantly upregulated in *AbMYB1*-overexpression lines compared to the wild type lines. However, there was no significant difference in the expression level of *AbF3*′*H*. Compared with the wild type, the expression levels of genes related to anthocyanin biosynthesis, such as *AbMYB1*, *AbF3H*, *AbF3*′*5*′*H*, *AbDFR* and *AbANS* in the fruits of *AbMYB1*-RNAi transgenic lines were significantly reduced, while there was no significant difference in that of *AbCHS*, *AbCHI*, *AbF3*′*H* and *Ab3GT* (Figure 7A–I).

In ripe fruits (25 days after fruit setting), compared to the wild type *A. belladonna*, there was no significant difference in the expression levels of *AbCHS*, *AbCHI*, *AbF3H*, *AbF3*′*H*, *AbF3*′*5*′*H*, *AbDFR*, and *AbANS* in the *AbMYB1*-overexpression transgenic plants, with only *AbMYB1* and *Ab3GT* expression levels increasing to around double, respectively. The expression levels of genes related to anthocyanin biosynthesis in the *AbMYB1*-RNAi transgenic lines were significantly reduced, including *AbMYB1*, *AbCHS*, *AbCHI*, *AbF3H*, *AbF3*′*5*′*H*, *AbDFR*, *AbANS*, and *Ab3GT* (Figure 8A–I). However, there was also no significant difference in the expression levels of *AbF3*′*H*, which was the same as in fruit setting for 13 days.

## 3. Discussion

Over the past few years, the regulatory mechanisms of anthocyanins have been well-studied in plants, and the MYB family factor is the most critical transcription factor in regulating anthocyanin biosynthesis [14], with R2R3-MYB transcription factors playing an important role in anthocyanin biosynthesis [15]. In this study, we identified an R2R3-MYB transcription factor, AbMYB1, related to anthocyanin biosynthesis in *A. belladonna*, which led to the formation of purple fruits.

During the biosynthesis of secondary metabolites, the biosynthesis genes and regulating factors always exhibit similar tissue distribution patterns to secondary metabolites [16,17]. For example, Wang et al. reported that AcMYB123 interacted with AcbHLH42 and spatiotemporally regulated anthocyanin biosynthesis specifically in the inner pericarp of kiwifruit (*Actinidia chinensis*) [18]. In wild-type *A. belladonna*, anthocyanin was mainly synthesized in the flowers and fruits, which was consistent with the tissue expression characteristics of *AbMYB1*. This suggests that AbMYB1 may be involved in anthocyanins synthesis. The expression profiles of *AbMYB1* almost coincided with the genes related to anthocyanin biosynthesis, including *AbCHS*, *AbCHI*, *AbF3H*, *AbF3*′*H*, *AbF3*′*5*′*H*, *AbANS*, *AbDFR,* and *Ab3GT,* during the development of *A. belladonna* fruits. Overexpression of *AbMYB1* in *A. belladonna* significantly increased the content of anthocyanin in the leaves, flowers, and fruits, while the anthocyanin content significantly reduced in the flowers and fruits of the *AbMYB1*-RNAi transgenic lines compared to the wild type, fully demonstrating the role of *AbMYB1* in the biosynthesis of anthocyanin in *A. belladonna.*

The R2R3 MYB subfamily is the largest group present in higher plants; it possesses a highly conserved DNA-binding domain containing up to two imperfect repeats, R2 and R3 [19]. The R2R3-MYB family has important functions in the regulation of secondary metabolism in many plants. In *Arabidopsis*, three R2R3-MYB proteins MYB11, MYB12, and MYB111, regulated the expression of the early biosynthetic genes *CHS*, *CHI,* and *F3H* [5], whereas R2R3-MYB TFs PAP1 and PAP2 activated the late biosynthesis enzymes such as *DFR* [20]. AbMYB1 and MYB transcription factors of other species have a high similarity in amino acid sequence; both have structural characteristics of R2R3-MYB transcription factors, including conserved R2 and R3 domains. Evolutionary analysis suggests that AbMYB1 was closely similar with SnMYB and PhDPL, and SnMYB positively correlated with the content of anthocyanin in fruit development [16]. PhDPL regulated the expression of various anthocyanin biosynthesis genes in *Petunia hybrida* [17]. These results suggest that AbMYB1 may work as a typical anthocyanin regulator.

In an MBW complex, an MYB subunit and bHLH subunit synergistically regulate anthocyanin synthesis by regulating the expression of different genes [14]. From the expression profiles of anthocyanin biosynthesis pathway genes in wild-type, *AbMYB1*-overexpression, and *AbMYB1*-RNAi transgenic plants, AbMYB1 had different regulatory effects on different genes of the anthocyanin biosynthesis pathway. In the leaves of *AbMYB1*-overexpression transgenic plants, the expressions of other genes except *AbCHI* and *AbCHS* were significantly up-regulated, indicating that the expressions of *AbCHI* and *AbCHS* may not be regulated by AbMYB1. In addition, in fruits growing for 13 days after setting, the critical period of anthocyanin synthesis, the expression levels of *AbCHI* and *AbCHS* genes in *AbMYB1*-RNAi transgenic plants were not significantly different from those of the wild type, while in fruits growing for 13 days after setting and 25 days after setting, *AbF3*′*H* gene expression was not significantly different from that of the wild type in *AbMYB1*-overexpression or *AbMYB1*-RNAi transgenic plants. This indicated that, during the regulation of anthocyanin biosynthesis, AbMYB1 may cross-talk with the other factors.

In the ripe fruits of *AbMYB1*-overexpression plants at 25 days after setting, only *AbMYB1* and *3GT* expressions increased significantly compared to wild-type plants. This might be due to the fact that AbMYB1 can regulate the expression of multiple metabolism pathway genes in *A. belladonna*, and similar results were found in the studies on the regulation of *Petunia* flower color by PhDPL [17]. However, in ripe fruits, anthocyanin content reached its highest peak, which may play a certain negative feedback effect on gene expression. A similar feedback mechanism has been reported in *LrAN2* expression, which could be caused by a high concentration of anthocyanin in *Lycium ruthenicum* fruits during fruit ripening [21]. The high expression of the *Ab3GT* gene was probably required to satisfy the flourishing period of anthocyanin glycosylation modification in the ripening period. This could be the reason why the expression of other genes related to the anthocyanin synthesis pathway was decreased in wild-type *A. belladonna* ripe fruits, while the expression of the *Ab3GT* gene was still increased.

## 4. Materials and Methods

### 4.1. Plant Materials

*A. belladonna* plants were cultivated in the plant garden of Southwest University, located in the City of Chongqing, China (29°45′ N, 106°30′ E). For germination, belladonna seeds were soaked in a GA solution (1mg/mL) for 2 days at room temperature, and then were surface-sterilized with 75% (*v*/*v*) ethanol for 1 min. Next, the seeds were surface-sterilized with NaClO solution containing 0.1% Triton for 15 min, and washed with sterile water f times. Lastly, the seeds were placed on a Murashige and Skoog (MS) medium for germination. Fully mature seeds were collected and utilized for germinating seedlings. The seedlings were cultivated on bacteria-free MS solid medium under controlled temperature conditions of 25 ± 1 °C and a photoperiod of 16 h of light followed by 8 h of darkness. Leaves from 4-week seedlings were employed for regenerating transgenic plants, following the procedures previously described in our laboratory [22,23].

### 4.2. Digital Gene Expression Pattern and Heat Map Analysis

The expression data (fragments per kilobase per transcript per million mapped reads, FPKM) of the anthocyanin biosynthesis pathway genes and *AbMYB1* were downloaded from the *A. belladonna* transcriptome database available at http://medicinalplantgenomics.msu.edu/ (accessed on 23 April 2024). The digital expression profiles were analyzed using MeV 4.9.0 software.

### 4.3. RNA Extraction and Gene Clone of MYB1

Total RNA from the *A. belladonna* fruits was extracted and reverse-transcribed to form cDNA using the FastQuant RT Kit (TianGen Biotech, Beijing, China). Primers for amplifying the full-length coding sequence (CDS) of AbMYB1 were designed using Vector NTI Suite 8.0. The CDS was amplified using the primers F-c*AbMYB1* and R-c*AbMYB1* (Appendix A). Specific primers were designed according to the sequence of AbMYB1 in the transcription database of *A. belladonna*, and high-fidelity enzymes were used to amplify AbMYB1 from the cDNA of wild-type *A. belladonna* fruit.

### 4.4. Isolation and Phylogenetic Analyses of AbMYB1 Gene

The *AbMYB1* gene was isolated and subjected to phylogenetic analyses to elucidate its evolutionary relationship. According to the analysis of the *A. belladonna* transcriptomic database, the R2R3 family MYB transcription factor *AbMYB1* with high expression in fruits was screened out.

The resulting amino acid sequences were compared to known functional MYB transcription factors using Vector NTI Suite 8.0. A phylogenetic tree was constructed using the Neighbor-Joining (NJ) method in MEGA 7.0. The cDNA sequence of *AbMYB1* was cloned into T vector (pJET1.2) for sequencing. According to the sequencing results, the cDNA sequence was analyzed [24].

### 4.5. Expression Analysis by Quantitative Real-Time PCR

Total RNA from different organs of *A. belladonna* was extracted and reverse-transcribed to cDNA using the FastQuant RT Kit (TianGen Biotech, Beijing, China). Quantitative real-time PCR (qRT-PCR) was performed using the iTaq^®^ Universal SYBR^®^ Green Supermix (Bio-rad) and the IQTM5 Multicolor Real-Time PCR Detection System (Bio-Rad, Hercules, CA, USA), and the 2^−ΔΔCT^ method was used to calculate the expression levels of the genes [25]. The phosphoglycerate kinase gene (*PGK*) was used as the internal reference gene [26]. Primers used for qRT-PCR are listed in Appendix A. All experiments were carried out with three biological repeats.

### 4.6. Determination of the Anthocyanin Content

Each sample with 0.1 g fresh plant material was placed and ground in a 10 mL Eppendorf tube, and then dissolved with a 2 mL methanol: HCl (99: 1, *v*/*v*) solution, mixed thoroughly, and then left overnight at 4 °C. Following this, 1.3 mL of ddH_2_O was added and mixed well, followed by the addition of an equal volume of chloroform. The mixture was inverted and mixed thoroughly and allowed to stand for a while. Subsequently, the water phase and the chloroform phase were separated by centrifugation at 10,000× *g* for 10 min. After centrifugation, the water phase was carefully collected, in which the anthocyanins were present, while chlorophyll remained in the chloroform phase. The absorption values of the water phase at 530 nm (A530) and 657 nm (A657) were measured using a UV spectrophotometer (UNICO, Shanghai, China). The relative anthocyanin contents were calculated using the formula: anthocyanin content = (A530–A657) unit/g FW (fresh weight). Each data point was derived from three biological replicates to ensure accurate and reproducible results.

### 4.7. Establishment of Transgenic A. belladonna Plants with AbMYB1 Overexpression or RNA Interference

The corresponding fragments of *AbMYB1* were selected to construct the overexpression plasmid and the RNA interference (RNAi) plasmid, respectively; pBI121-*AbMYB1* and pBIN19-i*AbMYB1*, were constructed following the methods described in our previous publication [27]. After confirmation by sequencing, pBI121-*AbMYB1* and pBIN19-i*AbMYB1* were separately introduced into *Agrobacterium tumefaciens* EHA105. Leaves from 4-week bacteria-free seedlings were used as explants for gene transformation. Genetic transformation was performed as described previously by Song and Walworth [28] with slight modifications. The regeneration media were liquid MS medium containing 0.5 mg·L^−1^ indole-3-acetic acid (IAA), 1 mg·L^−1^ zeatin, and 100 mg·L^−1^ kanamycin. Transgenic plants of *A. belladonna* were established and propagated according to the methods described previously [22,23]. The regenerated plantlets were planted on the substrates composed with vermiculite: pindstrap moss: perlite (6:3:1) and grown at 25 ± 1 °C with a photoperiodic conversion of 16 h light and 8 h dark. When the plantlets grew up to around 15 cm in height, transgenic lines were identified by PCR. Then they were transferred to a garden in Southwest University. Independent wild-type plants were used as the control (*n* = 8). For transgenic plants, independent *AbMYB1*-overexpression lines and independent *AbMYB1*-RNAi lines were randomly selected for further analysis, and there were three plant replicates for each line. Around two months later, when the plants entered the flowering and fruit period, plant phenotypes were characterized; meanwhile, the leaves, flowers, and fruits were harvested and used for gene expression and metabolite analysis.

## 5. Conclusions

The *AbMYB1* cloned and identified in this study was a fruit-expressed R2R3-MYB TF that positively regulated anthocyanin biosynthesis in *A. belladonna*. Overexpression of *AbMYB1* in *A. belladonna* significantly increased the expression of most anthocyanin biosynthesis pathway genes, thus promoting anthocyanin biosynthesis, while RNA interference with *AbMYB1* in *A. belladonna* resulted in significantly reduced expression of mostly anthocyanin biosynthesis pathway genes and anthocyanin content. These results provided new insights into the molecular mechanism of anthocyanin biosynthesis in *A. belladonna*, and probably provided a candidate gene for the molecular breeding of solanaceae plants to enhance anthocyanin content in fruits.

## Figures and Tables

**Figure 1 ijms-25-04963-f001:**
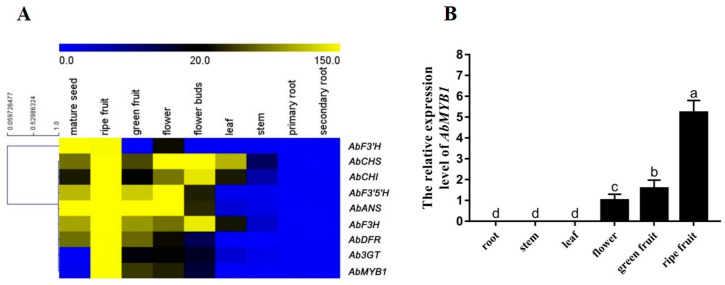
Expression characterization of *AbMYB1* and anthocyanin biosynthesis structural genes. (**A**) Digital gene expression pattern of *AbMYB1* and anthocyanin biosynthesis structural genes in *A. belladonna*. *AbF3*′*H*, Flavonoid 3′-hydroxylase; *AbCHS*, chalcone synthase; *AbCHI*, chalcone isomerase; *AbF3*′*5*′*H*, Flavonoid 3′, 5′-hydroxylase; *AbANS*, anthocyanidin synthase; *AbF3H*, flavanone 3-hydroxylase; *AbDFR*, dihydroflavonol 4-reductase; *Ab3GT*, UDP glucose flavonoid 3-O-glcosyl-transferase; *AbMYB1*, MYB1 transcription factor in *A. belladonna*. Heatmaps show relative expression of anthocyanin biosynthesis structural genes and *AbMYB1* in *A. belladonna.* (**B**) The transcriptional level of *AbMYB1* in different tissues of *A. belladonna*. The *AbPGK* gene was used as a reference gene for data normalization. Results are shown as mean ±SD (*n* =3). Values marked with the same letter within a sampling date are not significantly different at *p* <0.05 according to Duncan’s new multiple-range test; ANOVA, analysis of variance.

**Figure 2 ijms-25-04963-f002:**
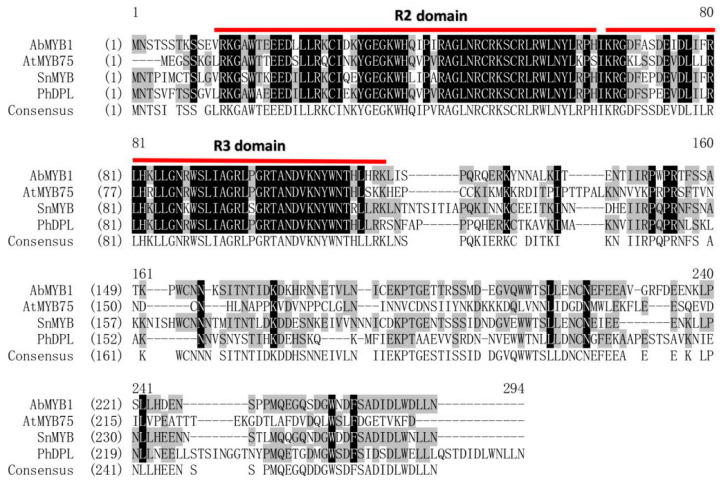
Multiple alignments of AbMYB1 with other plants regulating anthocyanin biosynthesis MYB. A C-terminal-conserved R2R3 motif is framed in the red line. The AbMYB1 protein sequence was used to query (via BLAST) the homologous proteins of different species such AtMYB75, SnMYB, and PhDPL. The same amino acids are shown in black on a white background, while similar amino acids are shown in black on a gray background.

**Figure 3 ijms-25-04963-f003:**
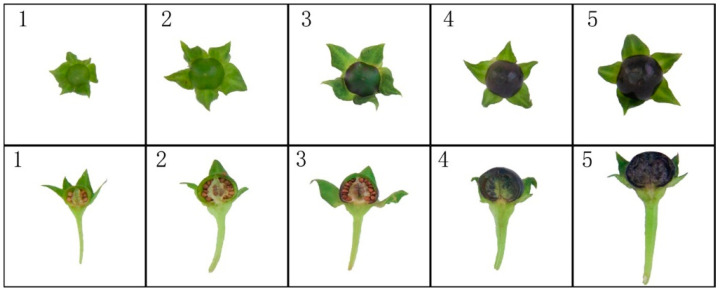
Phenotypic changes in wild-type *A. belladonna* fruits at different developmental stages: 1, young fruit (young fruit growing for 10 days from setting); 2, expansion-stage fruit (the fruit that entered the swelling stage after the young fruit grew for 15 days); 3, fruit of breaking stage (the fruit just turned purple); 4, purple fruit (purple fruit developed 5 days after color breaking); 5, ripe fruit (ripe fruit that developed for 10 days after color breaking).

**Figure 4 ijms-25-04963-f004:**
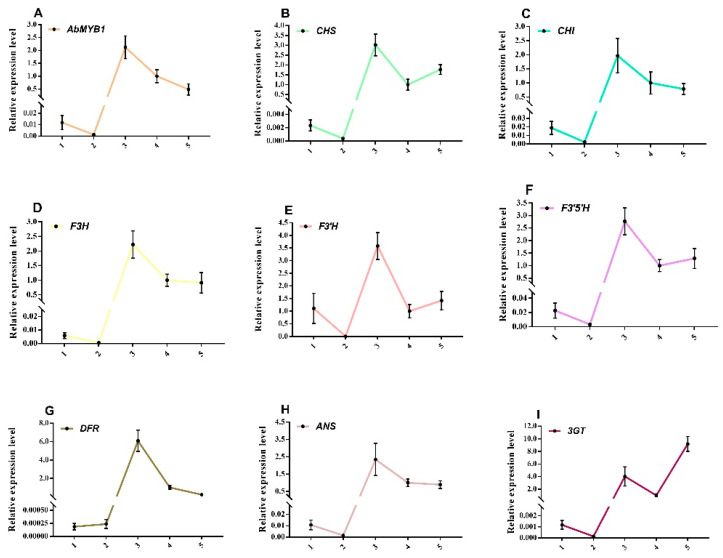
Expression profiles of anthocyanin biosynthesis-related genes in wild-type *A. belladonna* fruits at different developmental stages: 1, young fruit (young fruit growing for 10 days from setting); 2, expansion stage fruit (the fruit that entered the swelling stage after the young fruit grew for 15 days); 3, fruit of breaking stage (the fruit just turned purple); 4, purple fruit (purple fruit developed 5 days after color breaking); and 5, ripe fruit (ripe fruit that developed for 10 days after color breaking). (**A**) the expression level of *AbMYB1* at different stages; (**B**) the expression level of *AbCHS* at different stages; (**C**) the expression level of *AbCHI* at different stages; (**D**) the expression level of *AbF3H* at different stages; (**E**) the expression level of *AbF3*′*H* at different stages; (**F**) the expression level of *AbF3*′*5*′*H* at different stages; (**G**) the expression level of *AbDFR* at different stages; (**H**) the expression level of *AbANS* at different stages; and (**I**) the expression level of *Ab3GT* at different stages.

**Figure 5 ijms-25-04963-f005:**
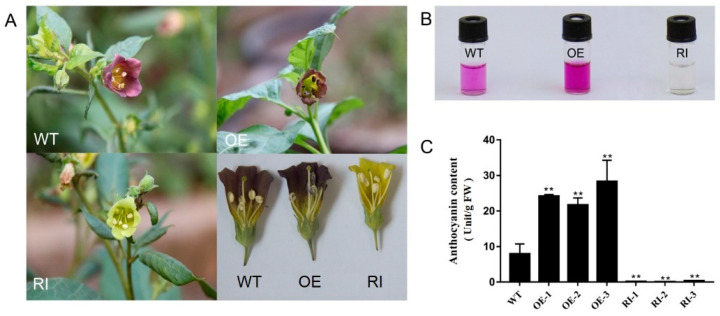
The anthocyanin accumulation in the flowers of transgenic *A. belladonna* plants. (**A**) The anthocyanin accumulation in the flowers of transgenic plant lines and wild-type plant lines; (**B**) total anthocyanin extracted from the flowers of transgenic plant lines and wild-type plant lines; (**C**) total anthocyanin content in the flowers of transgenic plant lines and wild-type plant lines. WT, the flowers of wild-type plants; *OE*, *OE-1*, *OE-2,* and *OE-3*, the flowers of *AbMYB1*-overexpression plants; *RI*, *RI-1*, *RI-2,* and *RI-3*, the flowers of *AbMYB1*-RNAi plants; ** means significant difference in T test (*p* < 0.01).

**Figure 6 ijms-25-04963-f006:**
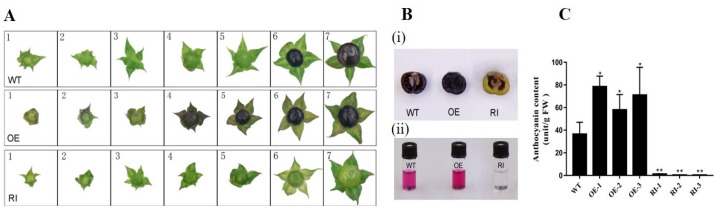
Anthocyanin accumulation in transgenic *A. belladonna* fruits at different developmental stages. (**A**) Phenotypic changes in transgenic *A. belladonna* fruits. WT, the fruits of wild type plants; OE, the fruits of *AbMYB1*-overexpression plants; RI, the fruits of *AbMYB1*-RNAi plants; 1, the fruit growing for 5 days from setting; 2, the fruit growing for 8 days; 3, the fruit growing for 11 days; 4, the fruit growing for 13 days; 5, the fruit growing for 15 days; 6, the fruit growing for 20 days; 7, the fruit growing for 25 days. (**B**) The phenotype and anthocyanin extraction of the ripe fruits of transgenic *A. belladonna.* i, the anthocyanin accumulation in ripe fruits of transgenic plant lines and wild-type plant lines; ii: total anthocyanin extracted from ripe fruits of transgenic plant lines and wild-type plant lines. (**C**) The anthocyanin accumulation in the ripe fruits of transgenic *A. belladonna.* Total anthocyanin content in ripe fruits of transgenic plant lines and wild-type plant lines. WT, the fruits of wild type plants; OE, the fruits of *AbMYB1*-overexpression plants; RI, the fruits of *AbMYB1*-RNAi plants; * means significant difference in T test (*p* < 0.05), ** means significant difference in T test (*p* < 0.01).

**Figure 7 ijms-25-04963-f007:**
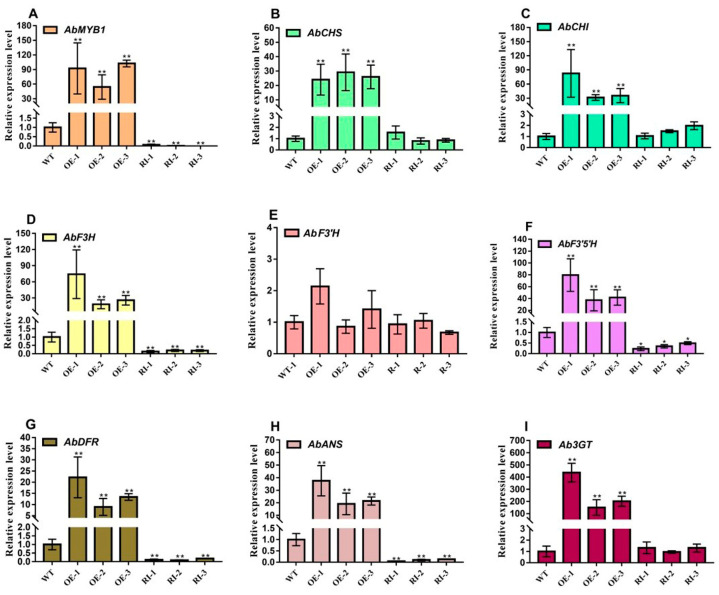
The relative expression of genes related to anthocyanin biosynthesis in *A. belladonna* fruits growing for 13 days after setting. (**A**) the expression level of *AbMYB1* in different lines; (**B**) the expression level of *AbCHS* in different lines; (**C**) the expression level of *AbCHI* in different lines; (**D**) the expression level of *AbF3H* in different lines; (**E**) the expression level of *AbF3*′*H* in different lines; (**F**) the expression level of *AbF3*′*5*′*H* in different lines; (**G**) the expression level of *AbDFR* in different lines; (**H**) the expression level of *AbANS* in different lines; (**I**) the expression level of *Ab3GT* in different lines; WT, the wild type plants; OE-1, OE-2 and OE-3: the *AbMYB1*-overexpression transgenic plant lines; RI-1, RI-2 and RI-3: the *AbMYB1*-RNAi transgenic pant lines; * means significant difference in T test (*p* < 0.05); ** means significant difference in T test (*p* < 0.01).

**Figure 8 ijms-25-04963-f008:**
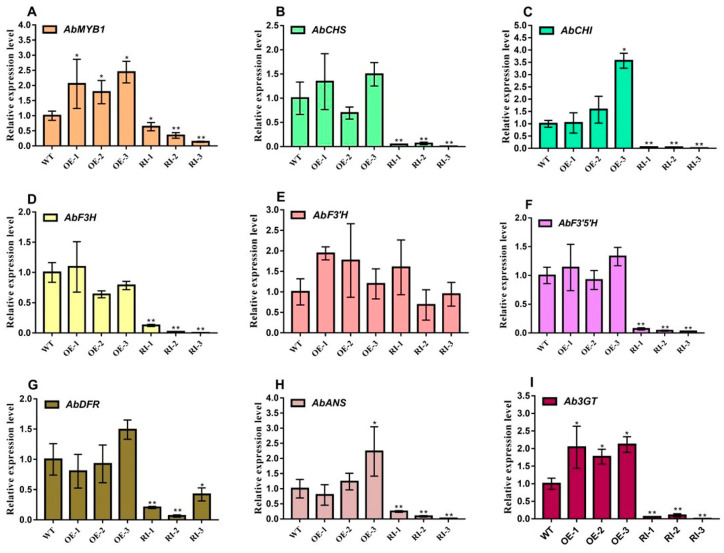
The relative expression of genes related to anthocyanin biosynthesis in *A. belladonna* ripe fruits at 25 days after setting. (**A**) the expression level of *AbMYB1* in different lines; (**B**) the expression level of *AbCHS* in different lines; (**C**) the expression level of *AbCHI* in different lines; (**D**) the expression level of *AbF3H* in different lines; (**E**) the expression level of *AbF3*′*H* in different lines; (**F**) the expression level of *AbF3*′*5*′*H* in different lines; (**G**) the expression level of *AbDFR* in different lines; (**H**) the expression level of *AbANS* in different lines; (**I**) the expression level of *Ab3GT* in different lines; WT, the wild type plants; OE-1, OE-2 and OE-3: the *AbMYB1*-overexpression transgenic plant lines; RI-1, RI-2 and RI-3: the *AbMYB1*-RNAi transgenic pant lines; * means significant difference in T test (*p* < 0.05); ** means significant difference in T test (*p* < 0.01).

## Data Availability

All the data obtained in this study have been presented in the manuscript, and no other data needs to be shared.

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
