# Peer review of "A Fruit-Expressed MYB Transcription Factor Regulates Anthocyanin Biosynthesis in Atropa belladonna"

_ijms, 2024, doi:10.3390/ijms25094963_

Round 1

Reviewer 1 Report

Comments and Suggestions for Authors

This article examines the molecular pathways of anthocyanin biosynthesis in the nightshade plant, Atropa belladonna. The findings presented are significant not only for advancing our understanding of fundamental mechanisms, but also for potential applications in enriching fruits with anthocyanins.

To enhance the relevance of this research to readers, the authors could expand the introductory section to provide a more detailed explanation of the importance of anthocyanins and potential areas of application, such as in the production of functional food, cosmetics, and medicine.

The article is well-structured and includes all necessary sections. The experimental section provides sufficient information on the research methodology to ensure reproducibility. The use of a significant amount of illustrative materials facilitates the comprehension of information by readers, while the clarity of figures aids in interpreting data. The research results in the article are not based on unreliable assumptions or extrapolations, which increases their reliability. The conclusions drawn from the experiments conducted are supported by the results obtained. The list of references provided in the article is comprehensive and relevant to the topic discussed. All cited sources are clearly identified within the text. I did not identify any instances of inappropriate self-citation.

I therefore recommend publication of the paper in journal.

Author Response

Dear Reviewer:

Thank you for your comments concerning our manuscript entitled “A fruit-expressed MYB Transcription Factor Regulates Anthocyanin Biosynthesis in Atropa belladonna” (ID: ijms-2974900). Those comments are all valuable and very helpful for revising and improving our paper, as well as the important guiding significance to our researches. We have studied comments carefully and have made correction which we hope meet with approval. Revised portions are marked in red in the paper. The main corrections in the paper and the responds to your comments are as following:

  1. Comment: To enhance the relevance of this research to readers, the authors could expand the introductory section to provide a more detailed explanation of the importance of anthocyanins and potential areas of application, such as in the production of functional food, cosmetics, and medicine.

Respond: Thank you very much for pointing out this important issue to improve our manuscript. According to your comment, we have added the introduction about the application of anthocyanins in introductory section, as follows: Due to the various colors and antioxidant capacity, anthocyanins are often used as color matching agent and preservative in the food industry. Besides, it’s also used in anti-aging, antioxidant, and whitening skincare products in the cosmetics industry, and treating cardiovascular diseases in the pharmaceutical industry. (line 46-50)

Reviewer 2 Report

Comments and Suggestions for Authors

The current manuscript is intriguing, but there are some major revisions that need to be addressed before acceptance

·       This research entitled A fruit-expressed MYB Transcription Factor Regulates Antho- 2 cyanin Biosynthesis in Atropa belladonna” has good academic significance. The topic addressed has good scientific depth.

·       They serve as attractants for animals, facilitating pollination and seed dispersal. How do anthocyanins contribute to pollination and seed dispersal in Plant. Explain in the abstract discussion part.

·       How was the R2R3-MYB transcription factor (TF), AbMYB1, identified in this study

·       Anthocyanin plays a very important role in the physiological processing of plants, for instance, attracting animals to pollinate, and spreading seeds" how anthocyanin attract animals to aid in pollination? Write the reason.

·       Write clear objective in the study to easily understand the reader.

·       Material and methods used in the research were not clear and explanatory. Write and explain in each section.

·       The first line in the Results section, which states (All the sequences of CHS, CHI, F3H, F3'H, F3'5'H, DFR, ANS, and 3GT were searched in the transcription database of A. belladonna, and Blastp analysis was conducted with the reported functional anthocyanin biosynthesis pathway genes in Solanaceae) should be omitted as it has already been mentioned in the Materials and methods section.

·       The section on "Isolation and phylogenetic analyses of AbMYB1 gene" should focus solely on describing the isolation and phylogenetic analyses of the AbMYB1 gene, without delving into the details of the materials and methods. Begin with a clear statement related to the heading, such as "The AbMYB1 gene was isolated and subjected to phylogenetic analyses to elucidate its evolutionary relationship.

·       The phylogenetic tree increase resolution and make beautiful.

·       The discussion section should be logically connected to the results by providing an interpretation and analysis of the obtained data. Additionally, it's important to compare the results with findings from other studies to provide context and validate the significance of the current findings. This comparison allows for a deeper understanding of the implications of the results and their contribution to the existing body of knowledge

·       Please cite and follow these papers.

1.     Umar Zeb, Azizullah Azizullah, Xiukang Wang, Sajid Fiaz, Hanif Khan. (2021). Comparative genome sequence and phylogenetic analysis of chloroplast for evolutionary relationship among Pinus species. Saudi Journal of Biological Sciences. https://doi.org/10.1371/ journal.

        doi: 10.1111/jse.12492

Comments on the Quality of English Language

Improved english

Author Response

Dear Reviewer:

Thank you for your comments concerning our manuscript entitled “A fruit-expressed MYB Transcription Factor Regulates Anthocyanin Biosynthesis in Atropa belladonna” (ID: ijms-2974900). Those comments are all valuable and very helpful for revising and improving our paper, as well as the important guiding significance to our researches. We have studied comments carefully and have made correction which we hope meet with approval. Revised portions are marked in red in the paper. The main corrections in the paper and the responds to your comments are as following:

  1. Comment: They serve as attractants for animals, facilitating pollination and seed dispersal. How do anthocyanins contribute to pollination and seed dispersal in plant. Explain in the abstract discussion part.

Respond: Thank you very much for pointing out this important issue to improve our manuscript. According to your comment, we have added the introduction about this in introductory section, as follows:

They serve as attractants for animals by providing plants with red, blue, and purple pigments, facilitating pollination and seed dispersal. (line 24-25)

Anthocyanin plays a very important role in the physiological processing of plants, for instance, attracting animals to pollinate, and spreading seeds by providing plants with red, blue, and purple pigments. (line 44-46)

  1. Comment: How was the R2R3-MYB transcription factor (TF), AbMYB1, identified in this study.

Respond: MYB1 was identified from A. belladonna transcriptomic database according to the gene expression pattern, and the related description was located in Chapter 4.4. (line 388-392)

  1. Comment: Anthocyanin plays a very important role in the physiological processing of plants, for instance, attracting animals to pollinate, and spreading seeds" how anthocyanin attract animals to aid in pollination? Write the reason.

Respond: Thanks for your suggestion, we added the reason why anthocyanin attract animals to aid in pollination. As follow: Anthocyanin plays a very important role in the physiological processing of plants, for instance, attracting animals to pollinate, and spreading seeds by providing plants with red, blue, and purple pigments. (line 44-46)

  1. Comment: Write clear objective in the study to easily understand the reader.

Respond: According to your comment, we have rewritten the objective of this study, as follow: In belladonna, both the skin and flesh of the fruit are rich in anthocyanins, reaching maximum abundance during the late ripening stage, and this is very rare in other Solanaceae plants. However, not much work has been carried out on the regulation of anthocyanins synthesis in belladonna genus fruits. (line 82-85)

  1. Comment: Material and methods used in the research were not clear and explanatory. Write and explain in each section.

Respond: According to your comment, we have added some information about Material and methods.

In Chapter 4.1. Plant materials. We had supplemented something about seed sterilization. The responds are as following: For germination, belladonna seeds were soaked in GA solution (1mg/ml) for 2 days at room temperature. And then were surface sterilized with 75% (v/v) ethanol for 1 min. Next, the seeds were surface sterilized with NaClO solution containing 0.1% Triton for 15 min, and washed with sterile water 5 times. At last, the seeds were placed on Mu-rashige and Skoog (MS) medium for germination. (line 364-369)

In Chapter 4.3. RNA extraction and Gene clone of MYB1. We had supplemented something about RNA extraction and gene clone. The responds are as following: Total RNA from A. belladonna fruits was extracted and reverse-transcribed to form cDNA using the FastQuant RT Kit (TianGen Biotech, Beijing, China). Primers for amplifying the full-length coding sequence (CDS) of AbMYB1 were designed using Vector NTI Suite 8.0. The CDS was amplified using the primers F-cAbMYB1 and R-cAbMYB1 (Appendix Table A1). Specific pri-mers were designed according to the sequence of AbMYB1 in the transcription data-base of A. belladonna, and high-fidelity enzymes were used to amplify AbMYB1 from the cDNA of wild type A. belladonna fruit. (line 380-387)

In Chapter 4.4. Isolation and phylogenetic analyses of AbMYB1 gene. We rewritten the analytical method. (line 388-392)

  1. Comment: The first line in the Results section, which states (All the sequences of CHS, CHI, F3H, F3'H, F3'5'H, DFR, ANS, and 3GT were searched in the transcription database of A. belladonna, and Blastp analysis was conducted with the reported functional anthocyanin biosynthesis pathway genes in Solanaceae) should be omitted as it has already been mentioned in the Materials and methods section.

Respond: According to your comment, we have deleted these words. (line 97)

  1. Comment: The section on "Isolation and phylogenetic analyses of AbMYB1 gene" should focus solely on describing the isolation and phylogenetic analyses of the AbMYB1 gene, without delving into the details of the materials and methods. Begin with a clear statement related to the heading, such as "The AbMYB1 gene was isolated and subjected to phylogenetic analyses to elucidate its evolutionary relationship.

Respond: According to your comment, we have divided this part into two chapters. The first chapter is Chapter 4.3 RNA extraction and Gene clone of MYB1. Total RNA from A. belladonna fruits was extracted and reverse-transcribed to form cDNA using the FastQuant RT Kit (TianGen Biotech, Beijing, China). Primers for amplifying the full-length coding sequence (CDS) of AbMYB1 were designed using Vector NTI Suite 8.0. The CDS was amplified using the primers F-cAbMYB1 and R-cAbMYB1 (Appendix Table A1). Specific pri-mers were designed according to the sequence of AbMYB1 in the transcription data-base of A. belladonna, and high-fidelity enzymes were used to amplify AbMYB1 from the cDNA of wild type A. belladonna fruit. (line 380-387)

The second chapter is Chapter 4.4 Isolation and phylogenetic analyses of AbMYB1 gene. The AbMYB1 gene was isolated and subjected to phylogenetic analyses to elucidate its evolutionary relationship. According to the analysis of A. belladonna transcriptomic database, the R2R3 family MYB transcription factor AbMYB1 with high expression in fruits was screened out. The resulting amino acid sequences were compared to known functional MYB transcription factors using Vector NTI Suite 8.0. A phylogenetic tree was constructed using the Neighbor-Joining (NJ) method in MEGA 7.0. The cDNA sequence of AbMYB1 was cloned into T vector (pJET1.2) for sequencing. According to the sequencing results, the cDNA sequence was analyzed. (line 388-397)

  1. Comment: The phylogenetic tree increase resolution and make beautiful.

Respond: According to your suggestion, we rebuild the phylogenetic tree, and the new image was shown in Appendix Figure A2.

  1. Comment: The discussion section should be logically connected to the results by providing an interpretation and analysis of the obtained data. Additionally, it's important to compare the results with findings from other studies to provide context and validate the significance of the current findings. This comparison allows for a deeper understanding of the implications of the results and their contribution to the existing body of knowledge.

Respond: According to your comment, we have revised the discussion section, and compared these results with the previous public reports to validate the significance of the current findings. The related content is located in line 301 to line 347.

  1. Comment: Please cite and follow these papers: Umar Zeb, Azizullah Azizullah, Xiukang Wang, Sajid Fiaz, Hanif Khan. (2021). Comparative genome sequence and phylogenetic analysis of chloroplast for evolutionary relationship among Pinus species. Saudi Journal of Biological Sciences. https://doi.org/10.1371/ journal.

Respond: We have cited this paper in Chapter 4.4. (line 397)

Reviewer 3 Report

Comments and Suggestions for Authors

Dear Authors,

In the manuscript „A fruit-expressed MYB Transcription Factor Regulates Anthocyanin Biosynthesis in Atropa belladonna“ transcription factor AbMYB1 was identified through association analysis of gene expression and anthocyanin accumulation in different tissues of Atropa belladonna. OE and RNAi lines were created and phenotypic analyses were performed. The expression pattern of AbMYB1 and 8 anthocyanin biosynthesis genes was followed in different organs and stages from plant development.

The manuscript is with high scientific relevance and provide novel information. It will be of interest for the readers. The Figure are informative and References are suitable.

I have some suggestions and comments which will be useful to improve the structure and vision of the manuscript:

1.      When you mention plant type for first time use the scientific name first and than point the common name.

2.      The name of the genes should be italicized, and the name of the proteins should not be italicized; please verify this throughout the manuscript

3.      In M&M protocol for seed sterilization is missing. In M&M 4.3. Isolation and phylogenetic analysis of AbMYB1 gene, the gene isolation and phylogenetic analysis should be separated in different subpoints.

4.      Appendix Figures and Tables should be in different file, not in the main text of the manuscript

5. Try not to comment on the results. Comments and suggestions should be made in the Discussion.

Author Response

Dear Reviewer:

Thank you for your comments concerning our manuscript entitled “A fruit-expressed MYB Transcription Factor Regulates Anthocyanin Biosynthesis in Atropa belladonna” (ID: ijms-2974900). Those comments are all valuable and very helpful for revising and improving our paper, as well as the important guiding significance to our researches. We have studied comments carefully and have made correction which we hope meet with approval. Revised portions are marked in red in the paper. The main corrections in the paper and the responds to your comments are as following:

  1. Comment: When you mention plant type for first time use the scientific name first and then point the common name.

Respond: According to your comment, we have corrected the name format of plant.

Atropa belladonna (Ab), (line 27)

Malus domestica (Md), (line 67)

Arabidopsis thaliana (At), (line 135)

Solanum nigrum (Sn), (line 135)

Petunia hybrida (Ph), (line 136)

  1. Comment: The name of the genes should be italicized, and the name of the proteins should not be italicized; please verify this throughout the manuscript.

Respond: According to your comment, we carefully examined the gene and protein name format throughout the manuscript, and correct those formatting errors. The gene names were italicized, and the protein names were without italicized.

  1. Comment: In M&M protocol for seed sterilization is missing. In M&M 4.3. Isolation and phylogenetic analysis of AbMYB1 gene, the gene isolation and phylogenetic analysis should be separated in different subpoints.

Respond: According to your comment, we have added the description about seed sterilization in Chapter 4.1, as follow: For germination, belladonna seeds were soaked in GA solution (1mg/ml) for 2 days at room temperature. And then were surface sterilized with 75% (v/v) ethanol for 1 min. Next, the seeds were surface sterilized with NaClO solution containing 0.1% Triton for 15 min, and washed with sterile water 5 times. At last, the seeds were placed on Murashige and Skoog (MS) medium for germination. (line 364-369)

According to your comment, Chapter 4.3 has been divided into two paragraphs, Chapter 4.3 and Chapter 4.4. (line 380-397)

  1. Comment: Appendix Figures and Tables should be in different file, not in the main text of the manuscript.

Respond: According to your comment, we adjust the position of Appendix Figures and Tables, and place these data in Appendix file.

  1. Comment: Try not to comment on the results. Comments and suggestions should be made in the Discussion.

Respond: According to your comment, we have revised the discussion section, and compared these results with the previous public reports to validate the significance of the current findings. The related content is located in line 301 to line 347.

Round 2

Reviewer 2 Report

Comments and Suggestions for Authors

I have meticulously reviewed the manuscript to ensure clarity, accuracy, and overall quality, and I am confident that it is well presented.

Author Response

Dear reviewer,

Thank you for your kindly comments to our manuscript and best wishes to you.

Reviewer 3 Report

Comments and Suggestions for Authors

Dear Authors,

Thank you very much about your answers. The manuscript has been improved. But still, there are some corrections you need to add. There are still omissions and remarks in the text that need to be corrected before acceptance:

1.           Please, add the scientific names for all the plants you mention in the text. Line 26 – add the scientific name latin name for tomato and eggplant; line 70 – add scientific name for rice - Oryza sativa (rice); line 74 – add scientific names for corn, apples and peaches

2.           The name of the genes/TFs should be italicized. Please, check this in the text. Line 29, 35, 88 - AbMYB1 in Italic “AbMYB1; Line 133, 134 - AbMYB1, SnMYB in Italic AbMYB1, SnMYB”, etc…..

3.           Line 381 - A. Belladonna in Italic “A. belladonna.

Author Response

Dear Reviewer,

Thank you for your comments concerning our manuscript entitled “A fruit-expressed MYB Transcription Factor Regulates Anthocyanin Biosynthesis in Atropa belladonna” (ID: ijms-2974900). The comments are all valuable and very helpful for revising and improving our paper, as well as the important guiding significance to our researches. We have studied comments carefully and have made correction which we hope meet with approval. Revised portions are marked in red in the paper. The main corrections in the paper and the responds to your comments are as following:

  1. Comment: Please, add the scientific names for all the plants you mention in the text. Line 26 – add the scientific name latin name for tomato and eggplant; line 70 – add scientific name for rice - Oryza sativa (rice); line 74 – add scientific names forcorn, apples and peaches.

Respond: According to your comment, we have added the scientific names for all the plants we mention in the text.

tomato (Solanum lycopersicum), and eggplant (Solanum melongena), (line 26)

Oryza sativa (rice), (line 71)

as pears (Pyrus ussuriensis) and peaches (Prunus persica) , (line 75)

Unlike O. sativa, Zea mays (corn), Malus pumila (apple) and P. persica, (line 75, 76)

  1. Comment: The name of the genes/TFsshould be italicized. Please, check this in the text. Line 29, 35, 88 - AbMYB1in Italic “AbMYB1; Line 133, 134 - AbMYB1, SnMYB in Italic “AbMYB1SnMYB”, etc…...

Respond: According to your comment, we carefully examined the gene and protein name format throughout the manuscript, and correct those formatting errors. When “AbMYB1” named as gene (nucleic acid) was italicized, while named as protein was not italicized.

AbMYB1 in Italic “AbMYB1, (line 90)

AbMYB1 in Italic “AbMYB1, (line 203, 204)

AbMYB1 not in Italic “AbMYB1, (line 339, 352)

AbMYB1 in Italic “AbMYB1, (line 386, 391, 393, 426, 427, 428, 429, 446)

  1. Comment: Line 381 - A. Belladonna in Italic “A. belladonna.

Respond: According to your comment, we have corrected it. (line 383)

Kind regards,

Xiaoqiang Liu